# Expression Mapping and Functional Analysis of Orphan G-Protein-Coupled Receptor GPR158 in the Adult Mouse Brain Using a GPR158 Transgenic Mouse

**DOI:** 10.3390/biom13030479

**Published:** 2023-03-05

**Authors:** Jinlong Chang, Ze Song, Shoupeng Wei, Yunxia Zhou, Jun Ju, Peijia Yao, Youheng Jiang, Hui Jin, Xinjin Chi, Ningning Li

**Affiliations:** 1Tomas Lindahl Nobel Laureate Laboratory, The Seventh Affiliated Hospital, Sun Yat-Sen University, Shenzhen 518107, China; 2The Clinical Oncology Department, The Seventh Affiliated Hospital, Sun Yat-Sen University, Shenzhen 518107, China; 3Department of Anesthesiology, The Seventh Affiliated Hospital, Sun Yat-Sen University, Shenzhen 518107, China; 4China-UK Institute for Frontier Science, Shenzhen 518107, China; 5The Fifth People’s Hospital of Datong City, Datong 038300, China

**Keywords:** GPR158, protein–protein interaction, brain regions, cell types, subcellular localization, 3D co-location, cerebral cortex

## Abstract

Aberrant expression of G-protein-coupled receptor 158 (GPR158) has been reported to be inextricably linked to a variety of diseases affecting the central nervous system, including Alzheimer’s disease (AD), depression, intraocular pressure, and glioma, but the underlying mechanism remains elusive due to a lack of biological and pharmacological tools to elaborate its preferential cellular distribution and molecular interaction network. To assess the cellular localization, expression, and function of GPR158, we generated an epitope-tagged GPR158 mouse model (GPR158^Tag^) that exhibited normal motor, cognitive, and social behavior, no deficiencies in social memory, and no anxiety-like behavior compared to C57BL/6J control mice at P60. Using immunofluorescence, we found that GPR158^+^ cells were distributed in several brain regions including the cerebral cortex, hippocampus, cerebellum, and caudate putamen. Next, using the cerebral cortex of the adult GPR158^Tag^ mice as a representative region, we found that GPR158 was only expressed in neurons, and not in microglia, oligodendrocytes, or astrocytes. Remarkably, the majority of GPR158 was enriched in Camk2a^+^ neurons whilst limited expression was found in PV^+^ interneurons. Concomitant 3D co-localization analysis revealed that GPR158 was mainly distributed in the postsynaptic membrane, but with a small portion in the presynaptic membrane. Lastly, via mass spectrometry analysis, we identified proteins that may interact with GPR158, and the relevant enrichment pathways were consistent with the immunofluorescence findings. RNA-seq analysis of the cerebral cortex of the GPR158^−/−^ mice showed that GPR158 and its putative interacting proteins are involved in the chloride channel complex and synaptic vesicle membrane composition. Using these GPR158^Tag^ mice, we were able to accurately label GPR158 and uncover its fundamental function in synaptic vesicle function and memory. Thus, this model will be a useful tool for subsequent biological, pharmacological, and electrophysiological studies related to GPR158.

## 1. Introduction

The G-protein-coupled receptors (GPCRs) are a class of seven transmembrane receptors involved in sensing environmental cues and transducing signals intracellularly via interactions with G proteins [1]. GPCRs are major mediators of signal transduction in the central nervous system and their members are the primary targets of certain FDA-approved drugs [2,3]. Nevertheless, the endogenous ligands of several GPCRs have not been identified, and these GPCRs are therefore referred to as orphan GPCRs [4].

GPCRs are responsible for regulating various cellular responses by binding to extracellular stimuli (ligands) such as hormones, growth factors, odors, or even photons [5,6]. They are essential components of the cell membrane [7] and consist of seven hydrophobic transmembrane α-helices that together form a well-folded, three-dimensional structure [8]. The main feature of GPCRs is the extracellular N-terminal domain with three extracellular loops (ECL) and three intracellular loops (ICL), and an eighth helix attached to the cell membrane at the C-terminal end (via a palmitoylation group) [9]. In response to extracellular stimuli, GPCRs cascade signals downwards through G proteins composed of α, β, and γ subunits [10]. Guanosine triphosphate (GTP) can be dephosphorylated to guanosine diphosphate (GDP) which is bound to the Gα subunit, resulting in the dissociation of Gα and GβGγ [11]. This dissociation activates each subunit of the G protein, enabling it to intervene in various downstream cell signaling pathways [5,11]. Hydrolysis of GTP to GDP restores the resting state of G proteins [5,11].

Previously identified to be associated with biological stress in a functional screen [12], GPR158 has been assigned to the glutamate-family GPCRs [13], although it is still considered an orphan receptor [14]. The distal end of the GPR158 C-terminus contains several phosphodiesterase E γ-like motifs and selectively recruits G proteins in the activated state [15]. In addition, the proximal portion of the C-terminus of GPR158 contains a conserved sequence that enhances the activity of the G-protein signaling regulator 7 (RGS7) [15].

Recently, GPR158 has come into focus among the research community. To date, studies on GPR158 have focused on Alzheimer’s disease (AD) [16,17], depression [4,14,18,19], tumors [13,20,21,22,23,24,25,26], and ocular hypertension [12,27,28]. Recently, using the Gene Expression Omnibus (GEO) database, researchers compared the temporal cortex of Alzheimer’s disease (AD) patients to cognitively normal samples, and identified GPR158 as one of the hub genes that is significantly differentially expressed and enriched in synaptic function associated with learning memory according to Gene Ontology (GO) and the Kyoto Encyclopedia of Genes and Genomes (KEGG) analyses [17]. In addition, GPR158, expressed on hippocampal CA3 neurons, mediates osteocalcin (OCN)-dependent regulation of hippocampal memory, in part through inositol-1,4,5-trisphosphate and brain-derived neurotrophic factors [29]. Furthermore, it was found that GPR158 is distributed in the postsynaptic membrane and is a binding partner of heparan sulfate proteoglycan (HSPG) glypican 4 (GPC4). Meanwhile, GPR158 is restricted to the proximal parts of the apical dendrites of CA3 that receive mossy fiber input [30].

Although these studies have investigated the molecular mechanism and function of GPR158 in several disease models, current understanding of GPR158 remains primitive due to a lack of fundamental knowledge of GPR158. For example, its localization at the tissue and cellular level has not been determined, very few GPR158-interacting proteins have been identified to establish a functional network, and effective antibodies and pharmacological tools are lacking, which impedes further investigation. It is thought that the precise location of GPR158 may largely determine its molecular function and even its contribution to the pathology and progression of various neurological and neuropsychiatric diseases. Although immunofluorescence staining using a GPR158 antibody on cultured cells works well [31], the results of immunofluorescence staining of brain tissue are poor and no individual cell morphology can be distinguished [32]. This prompted us to generate a GPR158^Tag^ mouse model in which GPR158 could be accurately labeled without affecting the mouse’s behavior. Accordingly, we attempted to elaborate the spatial localization of GPR158 at three levels—brain regions, cell types, and subcellular location (i.e., cerebral cortex)—and to delineate the molecular basis of GPR158 via various techniques such as immunofluorescence, mass spectrometry (MS),) RNA sequencing, behavioral tests, and 3D co-localization analysis. 

## 2. Materials and Methods

### 2.1. Generation of Gpr158-3×Flag-TeV-HA-T2A-tdTomato and Gpr158^−/−^ Mice

We inserted an epitope of 3×Flag-TeV-HA-T2A-tdTomato into the C-terminus of the endogenous GPR158 gene, and thus generated a mouse strain containing an allele of Gpr158-3×Flag-TeV-HA-T2A-tdTomato (GPR158^Tag^ mice for short) to allow tracking of GPR158. In addition, the mouse model could also be used to identify potential interacting proteins of GPR158 via affinity purification and MS techniques. Tobacco Etch Virus protease (TEVp) is a site-specific protease that cleaves proteins containing the amino acid sequence ENLYFQ|S/G [33]. In 2001, Donnelly et al. compared 2A sequences from four sources and found that the T2A sequence had the highest self-shearing rate, approaching 100% [34]. Indeed, the self-shearing peptide T2A can undergo a codon jump after transcription and start translating the next protein directly, allowing the two proteins located before and after T2A to be automatically separated. Consequently, translational products of tdTomato are separated from GPR158 and do not affect the configuration of GPR158. Hence, the HA or Flag tag can always trace and label GPR158 molecules even if they are transported to various locations distant from the soma, whereas tdTomato reflects the production of GPR158 and its cell of origin.

GPR158^Tag^ mice were produced using CRISPR/Cas9 and homologous recombination on a C57BL/6J background by Shanghai Model Organisms Company (Shanghai, China). The expression cassette of 3×Flag-TeV-HA-T2A-tdTomato was inserted into the locus at the termination codon of the Gpr158 gene (Ensembl: ENSMUSG00000045967). Cas9 mRNA, gRNA and the donor vector containing a 3-kb 5′ homologous arm, 3×Flag-TeV-HA-T2A-tdTomato and a 3-kb 3′ homologous arm were microinjected into zygotes of C57BL/6J mice to obtain F0 generation mice. F0 mice were back crossed to C57BL/6J mice to generate F1 generation mice, and the population was further expanded in this way. The offspring mice were genotyped using a polymerase chain reaction (PCR) assay with four primers (5′ to 3′): 

GGGAGAGCCAGAGAGGAGAA(WT-F),

GCCTCCTGGTGAATCTTGCT(WT-R),

CCCCTGGGAAATCCACAGTC (MU-F)

and GGATTCTCCTCGACGTCACC (MU-R).

Gpr158^−/−^ (Gpr158^tm1(KOMP)Vlcg^) mice were a gift from the Gerard Karsenty Lab at Columbia University (NY, USA). Mouse DNA was extracted from mouse tails for PCR using a Quick Genotyping Assay Kit for Mouse Tail purchased from Beyotime Biotechnology (Shanghai, China). The first two exons of Gpr158 were replaced with a LacZ cassette with a stop codon in Gpr158^−/−^ mice. The offspring mice were genotyped using PCR with four primers (5′ to 3′):

GTGTAGCCTCTGCCCACTTC(KO-wt-F),

CCTTTCTGTGCTTTCCTTGC(KO-wt-R),

CTGCTGGGGATGTAACCTGT(KO-tg-F),

ATCTCTCCTCTGCAGGACCA(KO-lacZ-R).

### 2.2. Animals and Behavioral Tests

Mice were maintained in a pathogen-free SPFII facility with controlled temperature (23 ± 1 °C) and relatively constant humidity (50 ± 10%) on a 12/12 h light/dark cycle (lights on at 07:00) in the Center of Laboratory Animal Science, Southern University of Science and Technology, China. For immunofluorescence staining, adult male mice at the postnatal day 60 (P60) were used, while male mice at the age of 2–3 months were used for behavioral tests. Mice were housed in a group of four to five in clear plastic cages with ad libitum access to drinking water and food. All experiments were conducted according to the NIH Guide for the Care and Use of Laboratory Animals (NIH publications no. 80-23, revised 1996), and the procedures were approved by the Animal Care Committee of the Southern University of Science and Technology, China.

### 2.3. Open Field Test (OFT)

An OFT paradigm was used to evaluate the phenotypes of locomotor activity and anxiety of the rodents [35]. Before the OFT, mice were given 30 min for acclimation. An open field box 40 cm long × 40 cm wide × 40 cm high was used. The field was equally divided into 16 squares, while the central 4 squares were set as the center area (20 cm × 20 cm). In the formal test, a mouse was put into the open field and monitored for 10 min. Traveled distance and time spent in each zone were recorded using EthoVision XT Version 10.0 software (Noldus Information Technology, Leesburg, VA, USA).

### 2.4. Three-Chamber Social Interaction Test

The rationale for the three-chamber social behavior test is based on the mouse’s natural tendency to live in groups and to explore new objects. The test was carried out to monitor the social activity occurring between two conspecifics, with each chamber being 20 cm long × 40 cm wide × 25 cm high. This test consisted of three stages: (1) Habituation: a mouse was put in the center chamber and had 5 min to explore the apparatus freely. Next, after the 5 min habituation, the mouse was gently caught and placed back into the center chamber, with both entrances closed using guillotine doors. (2) Sociability: one stranger mouse (S1) was put inside a wire cage next to a corner of a side chamber, while another wire cage with the same appearance and material was put in the symmetrically opposite position in the other side chamber with nothing inside it. The guillotine doors were then opened to allow the mouse to explore both side chambers for 10 min. (3) Social novelty: the experimental mouse was then guided into the center chamber, and another stranger mouse (S2) was put inside the empty cage. The experimental mouse then had 10 min to explore the whole apparatus. In this test, the mice S1 and S2 were conspecific at the same size and age [36,37,38,39]. The apparatus was cleaned with 75% ethanol between sessions. A wire cage with a diameter of 8 cm was used to restrain the test mice (i.e., S1 and S2), while a concentric circle with a diameter of 14 cm was set as an interacting area into which a test mouse would enter to interact with the test mouse inside the wire cage. The behaviors during the three phases were recorded and analyzed using Noldus EthoVision XT10 software (Wageningen, The Netherlands). The interaction time spent with Stranger 1 vs. Empty cage in the second phase was analyzed to indicate the mouse’s sociability, while the time spent with S1 vs. S2 in the third phase was used to indicate the social novelty index of the mouse. The social preference index was given by the ratio of the time spent interacting with unfamiliar conspecifics (Stranger 1; S1) to the total time the subject spent interacting with conspecifics (Stranger 1; S1) and objects (Empty) [37]. The social novelty index was given by the ratio of the amount of time spent with the novel conspecific (Stranger 2; S2) to the total amount of time spent with the familiar (Stranger 1; S1) and novel conspecifics (Stranger 2; S2) [37].

### 2.5. Novel Object Recognition Test

The novel object recognition test (NOR) was adapted from previously described experiments [40,41,42]. The procedure of the novel object recognition test was composed of two phases: familiarization and test. Mice had 30 min to become used to the environment and conditions before the formal behavioral test. In the phase of familiarization, each mouse was put into an open field box (40 cm long × 40 cm wide × 45 cm high) and allowed freedom to explore the area, in which two identical objects (#A and #B) were placed at a distance of 10 cm from each other. The mouse was taken out and put into its own home cage, and then re-exposed to the area, in which object #B had been replaced by another novel object, #C, for 5 min after 2 h. A zone of 3 cm around each object was set as the recognition area, while time spent in each recognition area was recorded using Noldus EthoVision XT10 software (Wageningen, the Netherlands). The apparatus was cleaned with 75% ethanol between sessions. The recognition index was calculated as Recognition Index (RI), RI = Tnovel/(Tnovel ^+^ Tfamiliar), where Tnovel and Tfamiliar represent the time spent in the recognition areas of the novel and familiar objects, respectively.

### 2.6. Immunofluorescence Staining

Immunofluorescence was adapted as previously described [43]. After anesthesia with isoflurane, mice were perfused transcardially with 4% PFA. Thereafter, the brains were fixed in 4% PFA at 4 °C overnight, cryoprotected, and dehydrated with 20% sucrose for one day and 30% sucrose for a second day. They were then embedded in Tissue-Tek OCT (Sakura Seiki, Nagano, Japan), frozen, and stored at −80 °C for cutting. Brains were sectioned at 30 μm. Targeted brain slices were selected and first rinsed three times with 1x PBS (10 min/time), and then blocked with 10% goat or donkey serum and 0.3% Triton X-100 in PBS (blocking solution) for 1 h at room temperature. The brain slices were then incubated with primary antibodies in blocking solution on a shaker at 4 °C overnight. The next day, slices were rinsed three times with PBS, and then incubated with secondary antibodies in blocking solution for 90 min. Subsequently, brain slices were washed with PBS for 10 min, and then incubated with 1 μg/mL DAPI for 10 min. Finally, brain slices were washed three times with PBS before being wet-mounted on a slide and coverslipped with fluoroshield mounting medium. Slides were imaged using a ZEISS LSM880 confocal microscope (Jena, Germany). Specific information about antibodies is provided in the Appendix A.

### 2.7. Fluorescent Images Quantification

#### 2.7.1. Localization and Quantification of tdTomato^+^ Neurons in Various Brain Regions in the Sagittal Plane of the Brain of Gpr158^Tag^ Mice Brains

For localization and quantification of tdTomato^+^ cells in the sagittal brain region of Gpr158^Tag^ mice, images were captured using a ZEISS LSM880 confocal microscope (Jena, Germany) at 10× magnification. The ZEISS software then stitched the captured images into a complete sagittal image of the whole brain. The images were acquired by selecting tdTomato^+^ channels of GPR158-expressing cells without bias. The images were smoothed and background-subtracted using ImageJ software. Afterwards, 2D point signals created using Imaris software (Bitplane) were rendered, individual tdTomato^+^-cell-rich brain regions were delineated according to point signal density, and each brain region was partitioned and color-rendered. The dot signal representing tdTomato^+^ cells was calculated using the image rendered by spots. The dot density of tdTomato^+^ cells in each brain region was calculated as the percentage ratio of the number of tdTomato^+^ cells in the observed brain region to the surface area of the same region. Point-density plot was produced for the raw stained data extracted using Imaris. Point-density plotting of tdTomato^+^ cells in each brain region was performed using the Point-density Shiny tools in Hiplot (https://hiplot.org, accessed on 17 September 2022), a comprehensive web platform for scientific data visualization.

#### 2.7.2. Co-Localization of Subcells and Synapses in 3D tdTomato^+^ Cells

For co-localization results of HA-labeled GPR158 spot signals and synaptophysin (SYN) or postsynaptic density protein-95 (PSD-95) spot signals, Z-stacked images (spacing 0.1 μm) were captured using a ZEISS LSM880 confocal microscope (Jena, Germany) at 40× or 63× magnification. Images were acquired by randomly selecting tdTomato^+^ channels and SYN^+^/PSD95^+^ channels without bias. The images were smoothed and background-subtracted using ImageJ software. Afterwards, a 3D volumetric surface rendering of each z-stack of tdTomato^+^ cells was created using Imaris software (Bitplane), while HA-antibody-labeled (Rabbit, 3724S, Cell signaling) GPR158 and the presynaptic marker SYN as well as the postsynaptic marker PSD95 were subjected to spot rendering. Using the MATLAB plug-in *Colocalize Spots*, co-localization analysis was performed for HA-antibody-labeled GPR158 with presynaptic marker SYN and postsynaptic marker PSD95, respectively (co-localization was determined when the radius of intersection of the two spots was greater than 0.24 μm). The distribution of HA-tagged GPR158 spots on and within the membrane of tdTomato^+^ cells was calculated using the MATLAB plug-ins *Split into Surface Objects* and *Compute Distance between Spots and Surfaces* in Imaris software (version 9.6.1).

### 2.8. Co-Immunoprecipitation and Protein Mass Spectrometry Identification

Male C57BL/6J wild-type and GPR158^−/−^ mice (five mice per group) were sacrificed at postnatal day 60 (P60). Cortical tissue was dissected from these mice and immediately frozen in liquid nitrogen. RNA extraction and the subsequent RNA-seq analysis were performed by Shanghai Zhongke New Life Biotechnology Co. Briefly, total RNA was isolated using TRIzol (Thermo Fisher Scientific, Shanghai, China) and 2 µg RNA per sample was used as an input material to prepare a cDNA library (paired-end 250 bp) for high-throughput sequencing using an Illumina HiSeq 2000 sequencing system. After filtering out low-quality and adapter sequences, reads were mapped to the mouse genome using Hisat2 (version 2.1.0) [44]. Reads were subsequently extracted using the Htseq-count script. Significantly differentially expressed genes were identified using volcano plots via the Volcano App by Jack Wang and Jianfeng in the Hiplot Team (hiplot.org, accessed on 17 September 2022). Gene set enrichment analysis (GSEA) (gsea-msigdb.org/gsea/index.jsp; accessed on 17 September 2022) was performed. The correlation coefficient grid chart was plotted using a free online platform for data analysis and visualization (bioinformatics.com.cn; accessed on 26 September 2022).

### 2.9. RNA Sequencing (RNA-seq)

Male C57BL/6J Wild-type and GPR158^−/−^ mice (five mice per group) were sacrificed at postnatal day 60 (P60). Cortical tissue was dissected from these mice and immediately frozen in liquid nitrogen. RNA extraction and the subsequent RNA-seq analysis were performed by Shanghai Zhongke New Life Biotechnology Co., Shanghai, China. Briefly, total RNA was isolated using TRIzol (Thermo Fisher Scientific, Shanghai, China) and 2 µg RNA per sample was used as an input material to prepare a cDNA library (paired-end 250 bp) for high-throughput sequencing using Illumina HiSeq 2000 sequencing system. After filtering out low quality and adapter sequences, reads were mapped to the mouse genome using Hisat2 (version 2.1.0) [44]. Reads were subsequently extracted using the Htseq-count script. Significantly differentially expressed genes were identified using volcano plots via the Volcano App by Jack Wang & Jianfeng in the Hiplot Team (hiplot.org, accessed on 17 September 2022). Gene Set Enrichment Analysis (GSEA) (gsea-msigdb.org/gsea/index.jsp, accessed on 17 September 2022) was performed. The correlation coefficient grid chart was plotted using a free online platform for data analysis and visualization (bioinformatics.com.cn, accessed on 17 September 2022).

### 2.10. Statistical Analysis

Independent-sample *t* tests were applied to study the differences among experimental data from the open field test, new object recognition test, and three-chamber socialization tests. Assumptions of normality and homogeneity of variance were studied using Shapiro–Wilk tests and Levene’s tests, respectively. The results in the graphs are expressed as mean ± standard error of the mean (SEM). A value of *p* < 0.05 was defined as statistically significant. IBM SPSS 25.0 was applied for statistical analysis. GraphPad Prism 8.0 was used to generate plots.

## 3. Results

### 3.1. Generation of GPR158^Tag^ Mice

We used CRISPR/Cas9 technology to obtain heterozygous GPR158^Tag^ mice with the GPR158 gene targeted by a knock-in epitope of 3×Flag-TeV-HA-T2A-tdTomato. Cas9 mRNA and gRNA were obtained via in vitro transcription. Subsequently, homologous recombinant vectors were constructed via In-Fusion cloning, and the final targeting vector contained a 3.0 kb 5′ homologous arm, 3×Flag-TeV-HA-T2A-tdTomato, and a 3.0 kb 3′ homologous arm (Figure 1A). Cas9 mRNA, gRNA, and donor vector were microinjected into the fertilized eggs of C57BL/6J mice to obtain F0-generation mice. PCR amplification and sequencing was used to identify positive F0-generation mice to be mated with C57BL/6J mice to obtain six positive F1-generation mice. Mice were genotyped using short-fragment PCR (Figure 1B). As shown by immunofluorescence staining, tdTomato^+^ cells represented cells capable of expressing GPR158, and HA-labeled dot signals represented GPR158 molecules (Figure 1C). This mouse line is subsequently referred to as the GPR158^Tag^ mouse model.

### 3.2. GPR158^Tag^ Mice Displayed No Deficits in Motor and Motor and Cognitive Behavior or Social Behavior and Social Memory, and No Anxiety-like Behavior

As we observed during husbandry, the homozygous and heterozygous GPR158^Tag^ mice showed healthy and normal development. To confirm this, wild-type mice and GPR158^Tag^ mice were compared after 2 weeks of adaptation to their new environment. Specifically, they were subjected to the open field test, the novel object recognition test, and the three-chamber test to verify whether GPR158^Tag^ mice had abnormalities in locomotion, anxiety, cognitive ability, or sociability and social memory (Figure 2A). The central area cumulative duration, average velocity, total distance of locomotion, and number of central area entries in GPR158^Tag^ mouse cohorts were similar to those in wild-type cohorts (central area cumulative duration, *p* = 0.6932; average velocity, *p* = 0.6095; total distance of locomotion, *p* = 0.6091; central area entries, *p* = 0.8111; two-tailed Student’s t test; Figure 2B,C), indicating that GPR158^Tag^ mice had no deficits in locomotor function and anxiety-like behavior. GPR158^Tag^ mice were then tested for cognitive ability and for sociability and social memory ability using the novel object recognition test and three-chamber test, respectively (*, *p* < 0.05; ***, *p* < 0.001; ns, not significant; two-tailed Student’s t test; Figure 2D–J). Social preference index values were calculated and assessed, and showed no difference in the sociability of GPR158^Tag^ mice compared to WT cohorts (Figure 2H; n = 12, WT, male; n = 7, GPR158^Tag^, male, P60-P90; two-tailed Student’s *t* test, *represents p-value less than 0.05; ns, not significant). Social novelty index values were also analyzed, suggesting no difference in the social novelty ability of GPR158^Tag^ mice compared to WT mice (Figure 2J; n = 12, WT, male; n = 7, GPR158^Tag^, male, P60-P90; two-tailed Student’s *t* test, * represents p-value less than 0.05; *** represents p-value less than 0.001; ns, not significant). Taken together, the results showed that GPR158^Tag^ mice did not exhibit cognitive impairment or abnormalities in social behavior or social novelty behavior in comparison to control mice.

### 3.3. Expression Pattern of tdTomato-Labeled GPR158^+^ Cells in Mouse Brain

We used immunofluorescence to detect the distribution of tdTomato-labeled GPR158^+^ cells expressed in the brain regions of GPR158^Tag^ mice. The results from sagittal sections showed that tdTomato-labeled GPR158^+^ cells were distributed in multiple brain regions with 13 of them displaying a denser GPR158 distribution in the slices (1.2 mm lateral to the midline), including the olfactory bulb (OB), cortex, hippocampus (Hip), caudate putamen (CPu), olfactory tubercle (Tu), central nucleus of the inferior colliculus (CIC), cerebellum, rostral periolivery region (RPO), parvicellular reticular nucleus (PCRt), parvicellular reticular nucleus alpha part (PCRtA), spinal trigeminal nucleus (Sp5), spinal trigeminal nucleus caudal part (Sp5C),and gelatinous layer of the caudal spinal trigeminal nucleus (Ge5; Figure 3A–C). It is of note that the distribution density of the GPR158^+^ cells was found to be substantially higher in several brain regions with smaller areas, such as PCRtA, PRO, Ge5, Sp5C, and Tu (Figure 3D). The 3D pie chart illustrates the percentage distribution of tdTomato-labeled GPR158^+^ cells in the above 13 brain regions, with greater percentage distributions in the cerebellum (34.08%), cerebral cortex (22.59%), CPu (15.08%), Tu (5.91%), OB (5.83%), and Hip (1.84%), suggesting that GPR158 may play an important role in these brain regions (Figure 3E). It has been reported that GPR158 is expressed in neurons in the CA3 region of the hippocampus [29], which is consistent with our results.

### 3.4. Specific Cell Types Expressing GPR158 in the Cerebral Cortex

To investigate whether GPR158 was expressed on oligodendrocytes, microglia, astrocytes, and/or neurons, we performed co-immunofluorescence analysis with cell-type-specific markers. Seeing as the cerebral cortex is one of the major brain regions in which GPR158 is functionally active, we first performed double staining using an mCherry antibody and an IBA1 antibody (i.e., to label microglia) in this region of adult GPR158^Tag^ mouse brains. We found that GPR158 was not expressed in microglia (Figure 4(a1–a4)). In addition, to investigate whether GPR158 was expressed in astrocytes, we co-stained brain slices using a GFAP antibody and found that GPR158 was not localized in astrocytes of the cerebral cortex of GPR158^Tag^ mice (Figure 4(b1–b4)).

Olig2 is a transcription factor that has been found to be involved in the phenotypic definition of oligodendrocytes [46]. Therefore, it has been often used as a marker for oligodendrocytes in mice. To identify whether GPR158 was expressed in oligodendrocytes in the cerebral cortex of GPR158^Tag^ mice, we performed tdTomato and Olig2 double staining and confirmed that GPR158 was not expressed in cortical oligodendrocytes (Figure 4(c1–c4)). Furthermore, double staining with an mCherry antibody and a NeuN (neuronal nuclear antigen) antibody in the cerebral cortex of GPR158^Tag^ mice revealed the presence of substantial overlapping fluorescent co-localization (Figure 4(d1–d4)).

To consolidate our findings, we harnessed another mouse model (GPR158^+/−^) in which beta galactosidase (β-gal) labels cells that express GPR158. We co-stained β-gal and various cell markers in the cerebral cortex of GPR158^+/−^ mice to locate GPR158 expression in specific cell types (Appendix A). The subsequent results revealed that GPR158 was only expressed in neurons in layer 2/3 of mouse cerebral cortex, in keeping with our above-described observations.

### 3.5. Specific Neuron Types with GPR158 Expression in the Cerebral Cortex of GPR158^Tag^ Mice

It is known that the alpha subunit of type II calcium-calmodulin-dependent protein kinase (Camk2a) is expressed in glutamatergic neurons in the neocortex, hippocampus and piriform cortex [47,48,49,50]. Parvalbumin (PV) is expressed in a subtype of interneurons with axonal projection. About 60% of PV^+^ cells were found to synaptically connect with pyramidal cells in layer 2/3 of the cortex in the dendritic peri- and proximal regions [51]. To investigate the specific types of neurons in which GPR158 was expressed in the cerebral cortex of GPR158^Tag^ mice, we co-stained tdTomato and Camk2a or PV on coronal sections, respectively. Immunofluorescence staining showed that GPR158 was expressed in neurons in layer 2/3 of the cerebral cortex in GPR158^Tag^ mice and tdTomato^+^ cells accounted for 15.52% of NeuN^+^ cells in the cerebral cortex of GPR158^Tag^ mice (Figure 5A,B). Immunofluorescence staining assays showed that tdTomato^+^ cells were co-stained extensively with Camk2a^+^ neurons in layer 2/3 cells of the cerebral cortex of GPR158^Tag^ mice, but rarely hardly ever with PV^+^ cells (Figure 5C). Collectively, in the cerebral cortex of GPR158^Tag^ mice, the majority of GPR158 was found to be expressed in Camk2a^+^ neurons, but rarely in PV^+^ neurons (Figure 5D, *p* < 0.0001).

### 3.6. Subcellular Localization of GPR158 in Cortical Layer 2/3 Neurons in GPR158^Tag^ Mice

To further determine the subcellular localization of GPR158 in the cerebral cortex of GPR158^Tag^ mice, we used Imaris software to build 3D surface models of individual tdTomato^+^ cells. The tdTomato^+^ cells were assessed to represent GPR158-expressing cells whilst the HA tag fused with GPR158 protein was used to trace GPR158 molecules. We performed single-cell modeling of tdTomato^+^ cells distributed in layer 2/3 of the cerebral cortex and counted the numbers of HA spot signals on the membrane and within the cytosol of individual tdTomato^+^ cells (Figure 6A,B, ***, *p* < 0.001. We found that GPR158 was distributed not only on the cell membrane but also significantly in the cytosol (Figure 6C; distribution of membrane HA, 26.36%; distribution of cytosolic HA, 73.64%).

To study the localization of GPR158 molecules at synapses, we used the HA tag to trace GPR158 molecules and considered the tdTomato^+^ cells to be GPR158-expressing cells, which were co-stained with SYN or PSD95, respectively. The surface and spot functions of Imaris software were used to create 3D cytostomes and 3D spots of individual cells (Figure 6D). Co-localization of the HA punctate signals with PSD95 or SYN punctate signals was assessed using the Matlab co-localization plug-in (Figure 6E). We found that GPR158 molecules labeled with HA bound more to PSD95 and less to SYN (Figure 6F,G, **, *p* < 0.01), which suggested that GPR158 was distributed abundantly at the synapse in the postsynaptic membrane and to a lesser extent in the presynaptic membrane. In addition, we attempted to use β-gal for subcellular localization of GPR158 molecules in the cerebral cortex of GPR158^+/-+^ mice. However, the subsequent results showed that β-gal was not able to trace GPR158 molecular localization but rather represented positive cells expressing GPR158, reminiscent of tdTomato^+^ cells in GPR158^Tag^ mice (Appendix A). Notably, tdTomato^+^ cells did not co-localize with IBA1^+^ cells; however, HA punctate signals were seen in IBA^+^ cells (Appendix A), implying that microglia might have engulfed GPR158 molecules when pruning synapses, although GPR158 was not indigenously expressed in microglia. This finding also implies that GPR158 had a localization at the synapse.

### 3.7. Identification and Analysis of GPR158 Interacting Proteins via Mass Spectrometry and RNA Sequencing

We envisaged that GPR158^Tag^ mouse model could serve as a sophisticated tool for not only tracing GPR158 but also identifying potential GPR158 interactors. RGS7 is one of the few known GPR158 interactors [15]. To identify novel GPR158 interactors, we performed a pull-down experiment using magnetic beads conjugated with an anti-Flag tag antibody to enrich GPR158 from brain tissues of GPR158^Tag^ mice and wild-type mice. We then screened candidate proteins that were likely to interact with GPR158 by comparing unique peptides of the brain proteomes of GPR158^Tag^ mice and wild-type mice using mass spectrometry (Appendix A). Interestingly, we found that these potential interacting proteins, except for RGS7 as a positive control, were not yet listed on the STRING database, nor reported in the literature (Figure 7A).

In addition, proteins that may interact with GPR158 are enriched in Gene Ontology Cellular Component (GOCC) pathways, including calcium and calmodulin-dependent protein kinase complexes, glutamate synapses and presynaptic active zone membranes, which coincided with our cell type localization and subcellular localization results (Figure 7B). Based on the results of the STRING database, we found that GPR158 and its potential interacting proteins were enriched in multiple synaptic-function-related pathways in mammalian phenotype ontology, such as decreased neurotransmitter release (MP:0003990) and abnormal miniature excitatory postsynaptic currents (MP:0004753; Figure 7C).

Furthermore, we performed RNA sequencing of cortical tissue from WT and GPR158^−/−^ mice to observe any possible transcriptional effect of GPR158 knockout on the GPR158 interactors. The interactor volcano plot showed significantly differentially expressed genes (*p*<0.05, |log_2_FC|=0.5) between WT mice and GPR158^−/−^ mice (Figure 7D). Next, we used the Pearson correlation coefficient to assess possible relationships between GPR158 and its interactors. Strikingly, we found that knockout of GPR158 profoundly strengthened the correlations among GPR158 interactors compared to the WT group (red lines represent positive correlations, green lines represent negative correlations, and thicker lines represent higher Pearson correlation coefficients) (Figure 7E,F). In addition, gene set enrichment analysis (GSEA) of the GPR158^−/−^ mouse cortical transcriptome revealed that GPR158 knockout significantly affected the chloride channel complex pathway in GOCC (NES (normalize enrichment score) = 1.541, *p* = 0.002) and also had a tendency to influence the intrinsic component of synaptic vesicle membrane pathway in GOCC (NES = 1.291, *p* = 0.054; Figure 7E,F).

## 4. Discussion

Multiple experimental findings suggest that GPR158 may play key functional roles in Alzheimer’s disease (AD) [16,17], depression [4,14,18,19], tumor formation [13,20,21,22,23,24,25,26], and intraocular pressure [12,27,28]. Here, we inserted the expression cassette of 3×Flag-TeV-HA-T2A-tdTomato into the site of the termination codon of the Gpr158 gene. Thereafter, we constructed a GPR158^Tag^ mouse model to trace GPR158 and used various techniques such as immunofluorescence, protein profiling, behavioral testing, and 3D reconstruction to delineate the spatial localization of GPR158 at three levels: brain regions, cell types, and subcellular localization. This is the first report to systematically define the spatial localization of GPR158 in the mouse brain using an epitope-tagged gene-targeted mouse model (Figure 1). We considered the question of whether the expression of 3×Flag-TeV-HA-T2A-tdTomato in GPR158^Tag^ mice would result in any abnormal behavior or the presentation of overt phenotypes in this model. To rule out this possibility, we performed several behavioral tests on GPR158^Tag^ male mice, including the open field test, the novel object recognition test, and the three-chamber sociability test. The open field test was used to examine the locomotor ability and anxiety levels of the mice [35]. The new object recognition test was used to test the ability of the mice to perceive novel objects [42]. Finally, the three-chamber sociability test was used to test the sociability and social memory of the mice; it is also a common test used to detect one of the core symptoms of autism (social deficits) [36]. Our behavioral findings showed that GPR158^Tag^ male mice did not present functional abnormalities in motor ability, anxiety, cognitive ability, or sociability and social memory (Figure 2). Considering the behavioral effects of estrogen in female mice [52], male GPR158^Tag^ mice were used in this study for the behavioral, immunofluorescence, and protein spectrum experiments.

In fact, we found that GPR158^+^ cells are distributed in several brain regions of the mouse brain, including the cerebellum, cerebral cortex, caudate putamen (striatum; CPu), hippocampus, olfactory tubercle, olfactory bulb, and many other brain regions. There is a large body of research showing that GPR158 ablation leads to a significant antidepressant phenotype [19] and deficits in spatial memory acquisition [53], which is thought to be inextricably associated with the function of two of the most intensively studied brain regions: the prefrontal cortex (PFC) [19] and hippocampus [29,30,53]. The interaction of GPR158 with RGS7 in the PFC of mice is a key mechanism leading to behavioral abnormalities in stress-induced depression [18]. The absence of GPR158 or RGS7 increased the excitability of neurons in PFC layer 2/3 and prevented the effects of stress [18]. Furthermore, it has also been shown that Gpr158 is expressed in neurons of the CA3 region of the hippocampus and transduces osteocalcin (OCN) regulation of hippocampal memory in part through inositol 1,4,5-trisphosphate and brain-derived neurotrophic factors [29]. Stylianos Kosmidis et al. showed that disruption of OCN/GPR158 signaling led to downregulation of the RbAp48 protein, mimicking the recognition memory deficit observed in the aging hippocampus [32]. Thus, these papers directly or indirectly indicate that GPR158^+^ cells are predominantly distributed in the cerebral cortex and hippocampus and may exert their function there, which is consistent with our observations of GPR158 expression distribution in the present study (Figure 3). However, our data also showed that GPR158 was highly expressed in several other brain regions including the cerebellum, caudate putamen, and olfactory bulb, which could be informative for future studies on novel functions of GPR158.

Using immunofluorescence, we confirmed that in the cerebral cortex of GPR158^Tag^ mice, GPR158 was expressed only in neurons, and not in astrocytes, microglia, or oligodendrocytes (Figure 4). Intriguingly, our results demonstrated that GPR158^+^ (i.e., tdTomato^+^) cells were mostly Camk2a^+^ cells, and a small number of PV^+^ cells, suggesting that the vast majority of GPR158 molecules are likely to be expressed in excitatory neurons, and rarely expressed in interneurons (Figure 5). Postsynaptic density protein-95 (PSD-95) is an important regulator of synaptic maturation that interacts with, stabilizes, and transports N-methyl-D-aspartate acid receptors (NMDARs) and α-amino-3-hydroxy-5-methyl-4-isoxazolepropionic acid receptors (AMPARs) to the postsynaptic membrane [54]. PSD-95 is associated with postsynaptic density rather than with the presynaptic membrane at brain synapses [55]. Overexpression of PSD-95 in hippocampal neurons promotes the maturation of glutamatergic synapses [56]. Expression of PSD-95 increases the aggregation and activity of postsynaptic glutamate receptors and the maturation of presynaptic terminals [56]. Thus, PSD95 is widely used as an excitatory postsynaptic localization marker. SYN is an integral membrane glycoprotein found in the presynaptic vesicles of neurons and in the corresponding vesicles of the adrenal medulla [57]. A study on GPR158–HSPG interaction showed that GPR158 was located in the postsynaptic membrane of the proximal segment of the CA3 apical dendrite that receives mossy fiber input [30]. We used these two synaptic markers to establish the specific synaptic localization of GPR158. Surprisingly, GPR158 was expressed not only in the postsynaptic membrane but also in the presynaptic membrane, being distributed more in the former and less in the latter (Figure 6). Further, we sought to establish the subcellular localization of GPR158 in the neuron soma. Strikingly, we found a greater abundance of GPR158 in the cytoplasm than at the cell membrane (Figure 6). It is well known that the G-protein-coupled receptors (GPCRs) are seven transmembrane proteins that form the largest single family of integral membrane receptors [58]. We then questioned why GPR158 was expressed preferentially in the cytoplasm. We postulated a scenario that required cytosolic GPR158 to be transported to distant synapses for neuronal activities. In addition, M. Elizabeth Fini et al. found that GPR158 was localized in the nucleus in prostate cancer [28]; however, we did not observe nuclear localization of GPR158 in healthy GPR158^Tag^ mice.

Regarding the molecular interaction and function of GPR158, several studies have reported a robust association between GPR158 and RGS7, which is considered a major interactor among the very few currently known interactors of GPR158 [15]. GPR158 can bind to homopolymers of RGS7 without causing their disassembly [15,59]. The complex between GPR158 and RGS7 was a key regulator of emotional behavior and had a critical role in controlling stress-induced changes in the excitability of a neuronal population. Deletion of GPR158 or RGS7 enhanced the excitability of layer 2/3 prefrontal cortex (PFC) neurons and blocked the effects of stress [18]. Our mass spectrometry analysis using GPR158^Tag^ mice identified 47 proteins that may interact with GPR158. We selected the 25 top-ranked candidate proteins, including Rgs7, Gnb5, Camk2g, Snap25, and Camk2a. We further analyzed them using the STRING database and against the transcriptomic profiling results of the cerebral cortex of GPR158^−/−^ mice (Figure 7), and the data suggested that GPR158 knockout might directly or indirectly strengthen the associations among Snap25, Camk2a, Camk2g, and other proteins that might interact with GPR158 (Figure 7).

Protein spectrum results indicated that GPR158 might have a direct or indirect interaction with Camk2a (Figure 7). Camk2a is exclusively found in glutamatergic pyramidal neurons in the cortex and hippocampus [49,60,61]. If GPR158 indeed interacts with Camk2a, this would be consistent with the immunofluorescence staining results showing GPR158 to be abundantly expressed in Camk2a^+^ neurons in the cerebral cortex. Forebrain-specific deletions of the Camk2a gene resulted in severe deficits in water maze and environmental fear learning, whereas mice with deletion restricted to the cerebellum had normal learning ability [62]. Furthermore, we found that temporal control deletion of the Camk2a gene in adult mice was no less detrimental to learning and synaptic plasticity than germline deletion [62]. In addition, transcriptomic results showed that knockout of GPR158 significantly affected the chloride channel complex, which includes the GABAα receptor; interestingly, the GABAα receptor can affect cognitive function. After contextual learning, GABAα-receptor-mediated potentiation of inhibitory synapses preceded excitatory synaptic plasticity, resulting in a decrease in the balance between excitatory and inhibitory (E/I) inputs at the synapse, which returned to pretraining levels within 10 min [63]. Therefore, it is likely that GPR158 affects cognitive and memory function through direct or indirect interaction with Camk2a.

Our mass spectrometry analysis revealed some promising GPR158 interactors (Figure 7), of which Snap25 might be interesting and relevant to neuropsychiatric studies. Snap25 was found to be an important component of the soluble NSF (N-ethylmaleimide-sensitive factor) attachment protein receptor complex (SNARE), which comprises synaptobrevin, syntaxin, and Snap25 [64] (Figure 7). It was reported that Snap25 not only participated in mechanisms of exocytosis involving calmodulin and SNARE proteins, but also triggered endocytosis in conjunction with the influx of voltage-dependent calcium channels via calcium microdomains [65]. Our RNA-seq results showed a tentative effect of GPR158 knockout on the intrinsic component of the synaptic vesicle membrane (Figure 7). Further, immunofluorescence staining showed that GPR158 was expressed on the presynaptic membrane. Thus, based on the above observations, it is likely that GPR158 may be involved in the release and recycling process of presynaptic membrane vesicles by interacting directly or indirectly with Snap25 in the presynaptic membrane.

Taken together, our data revealed localization of GPR158 at the cerebral subregional, cellular, and subcellular levels and provided new insights into the interactome of GPR158 using an epitope-tagged gene-targeted mouse model that could serve as a tool for reliable and high-throughput screening of novel GPR158 interactors.

## Figures and Tables

**Figure 1 biomolecules-13-00479-f001:**
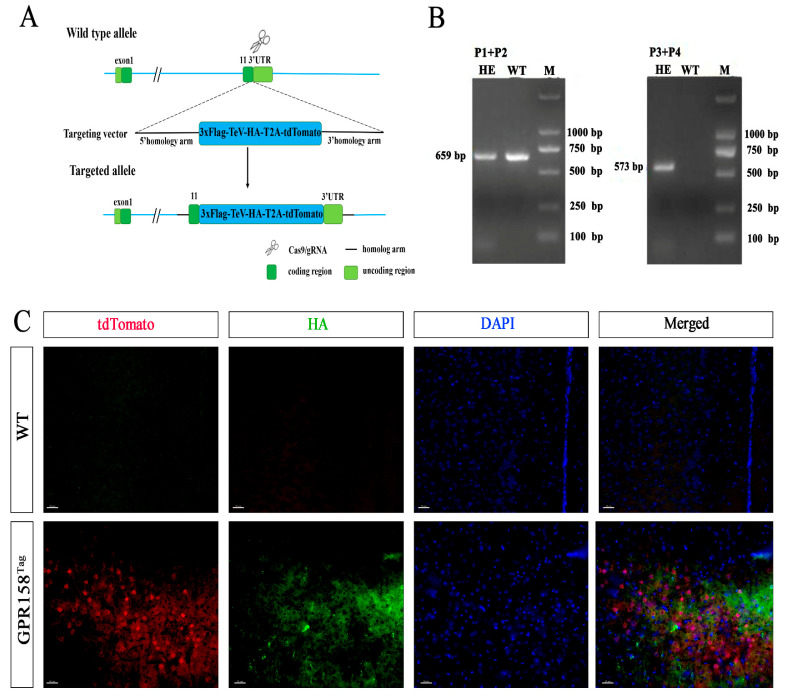
Generation of GPR158^Tag^ mice. (**A**) Schematic diagram of the GPR158^Tag^ mouse construction strategy. The 3×Flag-TeV-HA-T2A-tdTomato expression frame was targeted via homologous recombination at the Gpr158 gene stop codon site using CRISPR/Cas9 technology. Stably expressed Gpr158 knock-in 3×Flag-TeV-HA-T2A-tdTomato heterozygous mice were generated. (**B**) PCR genotyping analysis. Representative result of the PCR screening of GPR158^Tag^ mice showing 659 bp band (wild-type allele) and 573 bp band (mutant allele). (**C**) Immunofluorescence images of tdTomato and HA-labeled GPR158^+^ cells and GPR158 molecules, respectively, in the cerebral cortex of GPR158^Tag^ mice compared with Wild-type (WT) mice. Scale bars, 30 μm.

**Figure 2 biomolecules-13-00479-f002:**
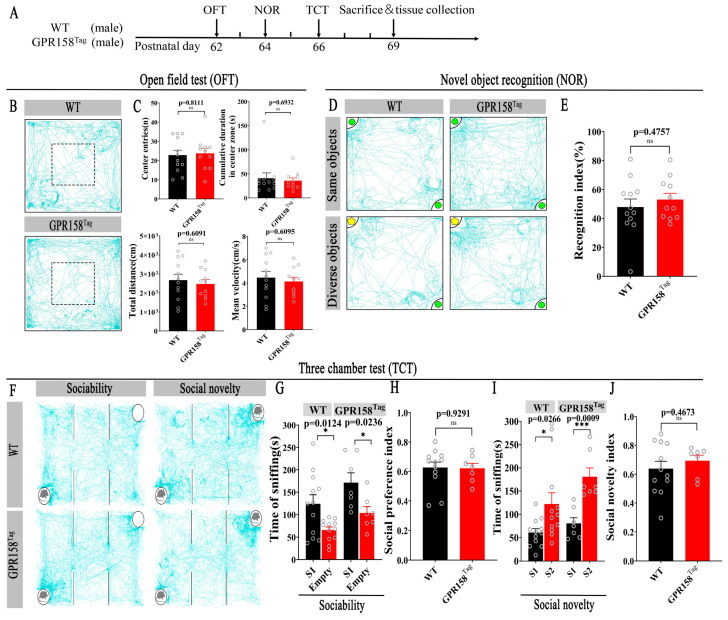
Behavioral tests of GPR158^Tag^ mice. (**A**) Timeline of behavioral tests of GPR158^Tag^ mice. (**B**) Representative trajectory diagrams interpreting time spent in different areas of the open field box for WT and GPR158^Tag^ mice. (**C**) Bar graphs showing absence of motor deficits and anxiety-like behavior in GPR158^Tag^ mice compared to control mice, as shown in the open field test (*n* = 12, WT, male; *n* = 11, GPR158^Tag^, male, P60-P90; ns, not significant; two-tailed Student’s *t* test). (**D**) Representative trajectory diagrams illustrating exploration time of WT and GPR158^Tag^ mice for familiar and novel objects during novel object recognition test. (**E**) Bar graphs showing no cognitive deficits in GPR158^Tag^ mice compared to control mice during the novel object recognition experiment (*n* = 12, WT, male; *n* = 11, GPR158^Tag^, male, P60-P90; *p* = 0.4757, two-tailed Student’s *t* test). (**F**) Representative trajectory diagrams interpreting the duration of socialization of WT and GPR158^Tag^ mice with Stranger Mouse 1 and the duration of novelty of socialization with Stranger Mouse 2 in the three-chamber test. (**G**,**H**) GPR158^Tag^ mice showed normal sociability compared to WT mice in the three-chamber socialization experiment (**G**). Social preference index was calculated so that it could be judged that there was no difference in the sociability of GPR158^Tag^ mice compared to WT mice (**H**); *n* = 12, WT, male; *n* = 7, GPR158^Tag^, male, P60-P90; two-tailed Student’s *t* test, * represents *p*-value less than 0.05; ns, not significant). (**I**,**J**) Three-chamber sociability test demonstrating no social novelty deficits in GPR158^Tag^ mice compared to WT mice (**I**). Social novelty index was calculated so that it could be judged that there was no difference in the social novelty ability of GPR158^Tag^ mice compared to WT mice (**J**); *n* = 12, WT, male; *n* = 7, GPR158^Tag^, male, P60-P90; two-tailed Student’s *t* test, * represents p-value less than 0.05; *** represents *p*-value less than 0.001; ns, not significant). Data are mean ± SEM.

**Figure 3 biomolecules-13-00479-f003:**
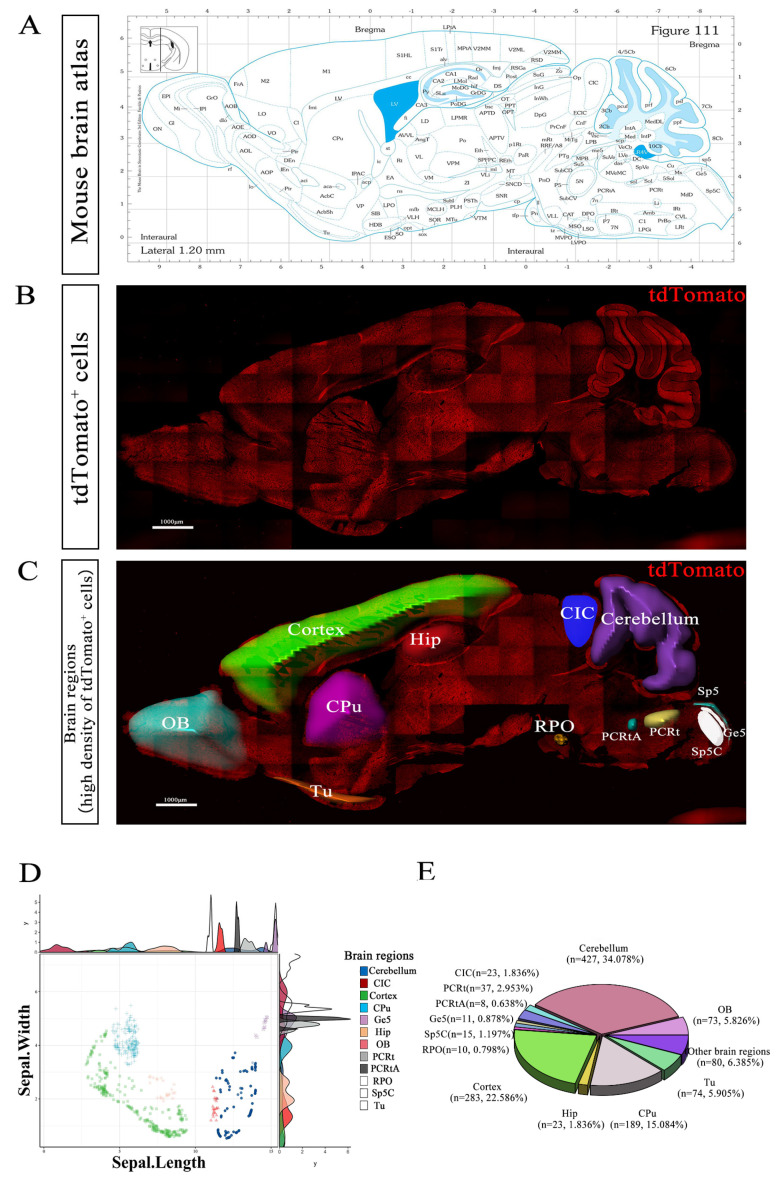
Distribution of tdTomato^+^ cells in multiple brain regions of GPR158^Tag^ mice at P60. (**A**) Brain atlas of mice with a sagittal section (1.20 mm lateral to the midline), adapted from *Paxinos and Franklin’s The Mouse Brain in Stereotaxic Coordinates*) [45]. (**B**) Sagittal section of the brain of a GPR158^Tag^ mouse (1.20 mm lateral to the midline) showing that tdTomato-tagged GPR158 was detected using immunofluorescence staining with anti-mCherry antibody. (**C**) The 13 brain regions with relatively high densities of tdTomato^+^ cells in the GPR158^Tag^ mouse brain sagittal section shown in panel B, annotated using Imaris software. (**D**) Dot plot combined with density plot showing the distribution and density of tdTomato^+^ cells in each of the 13 brain regions. (**E**) 3D pie chart showing the percentage distribution of tdTomato+ cells in 13 brain regions (OB, olfactory bulb; cortex; Hip, hippocampus; CPu, caudate putamen; Tu, olfactory tubercle; CIC, central nucleus of the inferior colliculus; cerebellum; RPO, rostral periolivery region; PCRt, parvicellular reticular nucleus; PCRtA, parvicellular reticular nucleus, alpha part; Sp5, spinal trigeminal nucleus; Sp5C, spinal trigeminal nucleus, caudal part; Ge5, gelatinous layer of the caudal spinal trigeminal nucleus). Scale bars, 1000 μm.

**Figure 4 biomolecules-13-00479-f004:**
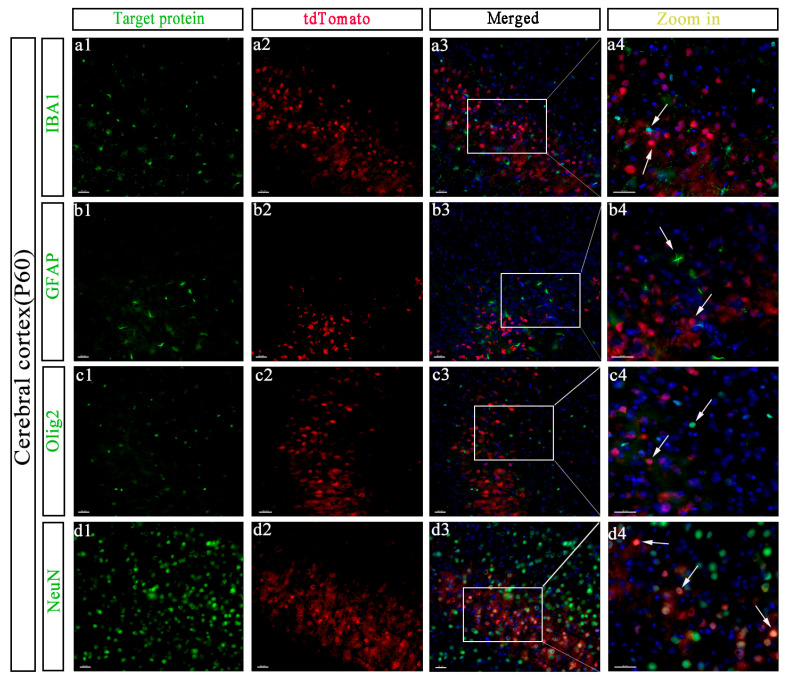
Expression of GPR158 in neurons in the cerebral cortex of Gpr158^Tag^ mice at P60. (**a1–a4**) Immunofluorescence staining on cerebral cortex showed GPR158 (red) was not expressed in microglia. Microglia (green, **a1**) were stained with anti-IBA1 antibody. Local zoomed-in diagram showing the localization of tdTomato^+^ cells in relation to IBA1^+^ cells in the cerebral cortex of GPR158^Tag^ mice at P60. (**b1–b4**) GPR158^+^ cells were not detected in activated astrocytes (green) of the cerebral cortex of GPR158^Tag^ mice at P60. Activated astrocytes (green, **b1**) were stained with anti-GFAP antibody. Local zoomed-in diagram showing the localization of tdTomato^+^ cells in relation to GFAP^+^ cells in the cerebral cortex of GPR158^Tag^ mice at P60. (**c1–c4**) GPR158 was not expressed in oligodendrocytes (green, **c1**). Oligodendrocytes (green) were stained with anti-olig2 antibody. Local zoomed-in diagram showing the localization of tdTomato^+^ cells in relation to olig2^+^ cells in the cerebral cortex of GPR158^Tag^ mice at P60. (**D**) GPR158 (red, **d1–d4**) were expressed in neurons (green, **d1**) in the cerebral cortex of GPR158^Tag^ mice at P60. Neurons (green) were stained with anti-NeuN antibody. GPR158 (red) were stained with anti-mCherry antibody. Nuclei (blue) were stained with DAPI. Scale bars, 30 μm.

**Figure 5 biomolecules-13-00479-f005:**
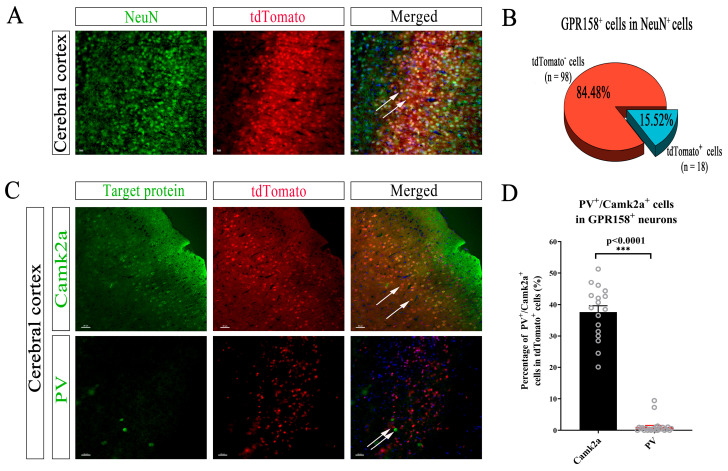
Cell types expressing tdTomato^+^ neurons in the cerebral cortex of Gpr158^Tag^ mice at P60. (**A**) Immunofluorescence staining with an anti-NeuN antibody to label neurons and an anti-mCherry antibody to label tdTomato^+^ cells expressing GPR158. (**B**) Percentage of tdTomato^+^ cells among NeuN^+^ cells in the cerebral cortex of Gpr158^Tag^ mice at P60 (12 slices from Gpr158^Tag^ mice (*n* = 3, male)). (**C**) Immunofluorescence staining showing excitatory neurons labeled with an anti-Camk2a antibody and inhibitory neurons labeled with an anti-PV antibody, respectively. These cells were co-stained with tdTomato^+^ neurons in layer 2/3 of the cerebral cortex of Gpr158^Tag^ mice.(**D**) Percentage of double-positive cells (i.e., tdTomato^+^ neurons co-stained with PV^+^ or Camk2a^+^ neurons) in total tdTomato^+^ neurons in the cerebral cortex of Gpr158^Tag^ mice (PV^+^, 25 slices; Camk2a^+^, 17 slices from Gpr158^Tag^ mice (*n* = 3, male, P60); ***, *p* < 0.0001). Scale bars, 30 μm.

**Figure 6 biomolecules-13-00479-f006:**
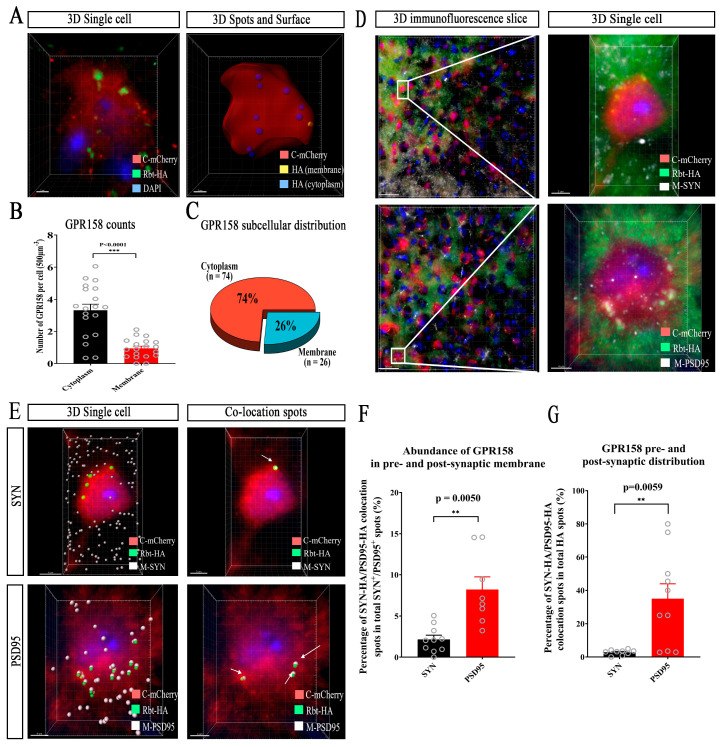
Analysis of subcellular localization of GPR158 in cortical layer 2/3 neurons in GPR158^Tag^ mice at P60. (**A**) 3D model of GPR158 and its distribution on the membrane and in the cytosol of individual neurons in layer 2/3 of the GPR158^Tag^ mouse cerebral cortex using HA-labeled GPR158 and tdTomato-labeled cell bodies. (**B**) Abundance of HA-labelled GPR158 on the membrane and in the cytosol of individual neurons in layer 2/3 of the cerebral cortex of GPR158^Tag^ mice (tdTomato^+^ cells (n = 19) from GPR158^Tag^ mice (n = 4, male, P60), ***: *p* < 0.001). (**C**) Percentage of HA-labeled GPR158 distributed on the membrane and in the cytosol of individual neurons in layer 2/3 of the cerebral cortex of GPR158^Tag^ mice (tdTomato^+^ cells (n = 19) from GPR158^Tag^ mice (n = 4, male, P60), ***: *p* < 0.001). (**D**) Immunofluorescence staining showing co-localization of HA-labeled GPR158 with SYN or PSD95 in the cerebral cortex of GPR158^Tag^ mice at single cell resolution. (**E**–**G**) 3D co-localization of HA-labeled GPR158 with SYN (**E**) or PSD95 (**E**) in individual cells in the cerebral cortex of GPR158^Tag^ mice, the percentage (**F**) of GPR158 binding to SYN or PSD95, and the pre- and postsynaptic distribution (**G**) of HA-labeled GPR158 (SYN^+^ cells: n = 10; PSD95^+^ cells: n = 10; GPR158^Tag^: n = 4, P60, male; **: *p* < 0.01). Data are mean ± SEM.

**Figure 7 biomolecules-13-00479-f007:**
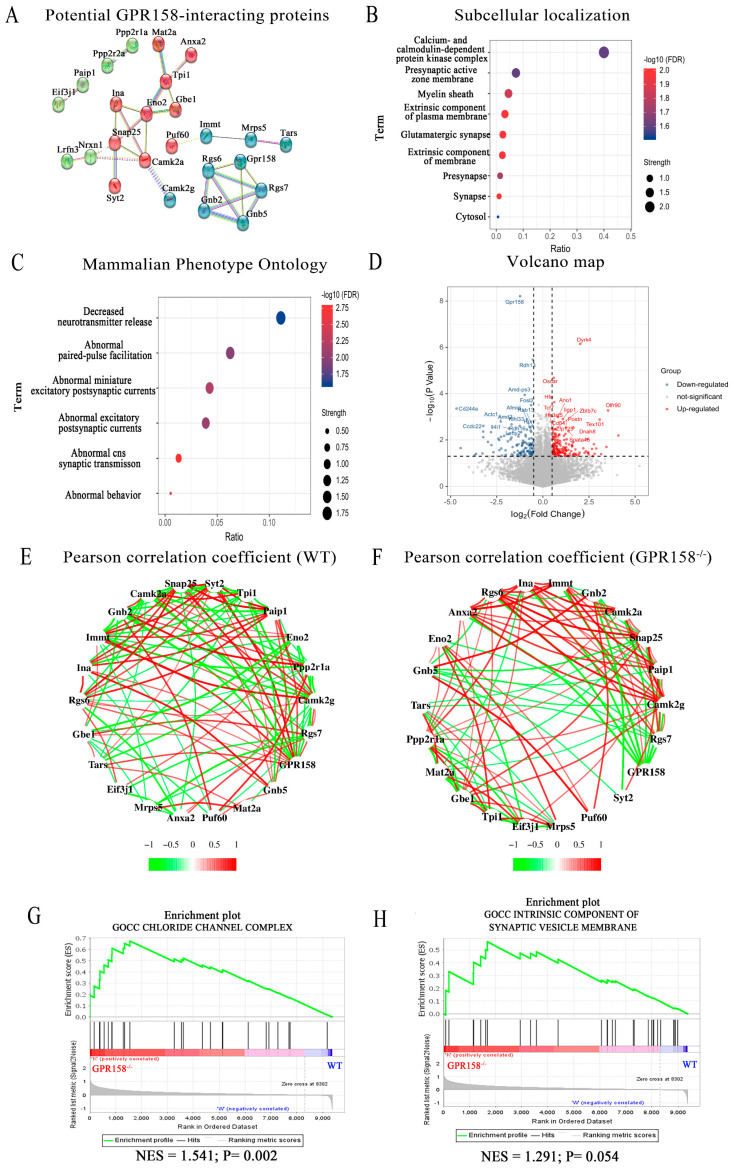
Identification and analysis of GPR158 interactors that displayed altered linear correlations and signaling pathways upon knockout of GPR158. (**A**) A network of proteins that may interact with GPR158 according to mass spectrometry analysis, established using a STRING database. According to the k-means algorithm, the proteins that may interact with GPR158 were classified into three clusters (WT mice, n = 3; GPR158^Tag^ mice, n = 3; male; See Appendix A for the screening process used to identify proteins that may interact with GPR158). (**B**) Analysis of subcellular localization of GPR158 and its potential interacting proteins obtained from GPR158^Tag^ mouse protein spectrum data using the STRING database. (**C**) Bubble plot showing that GPR158 and its potential interacting proteins were enriched in multiple synaptic-function-related pathways in mammalian phenotype ontology. (**D**) Volcano plot revealing significantly differentially expressed genes in the cerebral cortex transcriptomes of GPR158^−/−^ mice vs. WT mice (WT mice, n = 5; GPR158^−/−^ mice, n = 5; male; criteria for determining significantly differentially expressed genes, *p* < 0.05, |log_2_FC| = 0.5). (**E**,**F**) A Pearson correlation coefficient network of proteins that may interact with GPR158 in the cerebral cortex of WT (**E**) and GPR158^−/−^ (**F**) mice. Pearson correlation coefficient analysis was performed using FPKM (fragments per kilobase million) values from the RNA sequencing of WT (**E**) and GPR158^−/−^ (**F**) mice (WT mice, n = 5; GPR158^−/−^ mice, n = 5; male; red lines represent positive correlations, green lines represent negative correlations, and thicker lines represent larger Pearson correlation coefficients). (**G**,**H**) Gene set enrichment analysis (GSEA) of genes enriched in Gene Ontology Cellular Component pathways in the cortical RNA sequences of WT and GPR158^−/−^ mice (G: GPR158^−/−^ vs. WT, NES (normalized enrichment score) = 1.541, *p* = 0.002; H: GPR158^−/−^ vs. WT, NES= 1.291, *p* = 0.054; WT mice, n = 5; GPR158^−/−^ mice, n = 5; male).

## Data Availability

All data needed to evaluate the conclusions in the paper are present in the paper. The preclinical data sets generated and analyzed during the current study are not publicly available but are available from the corresponding author upon reasonable request. The data will be provided following the review and approval of a research proposal with a statistical analysis plan and execution of a data sharing agreement. The data will be accessible for 12 months for approved requests, with possible extensions considered. For more information on the process or to submit a request, contact the corresponding authors.

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
