# Peer review of "Expression Mapping and Functional Analysis of Orphan G-Protein-Coupled Receptor GPR158 in the Adult Mouse Brain Using a GPR158 Transgenic Mouse"

_biomolecules, 2023, doi:10.3390/biom13030479_

Round 1
Reviewer 1 Report
In the manuscript entitled Expression mapping and functional analysis of orphan 2 G-protein coupled receptor GPR158 in the adult mouse brain 3 using a GPR158 reporter mouse the authors describe the localization of the orphan receptor GPR158 in the CNS by generating an epitope-tagged GPR158 mouse. They describe that the receptor is only expressed in neurons of the cerebral cortex, hippocampus, cerebellum and caudate putamen. Finally, the authors demonstrate the possible interaction of GPR158 with 12 other proteins
involved in the chloride channel complex and synaptic vesicle membrane composition, becoming an important target for synaptic vesicle function and memory. The manuscript offers new data, it is clear and well written. However there are some minor points that could be improved:
Figure 3A,B contain really small information that can not be read. It would be nice if the authors could create a new Figure or change the distribution.
Figure 4. GFAP and olig2 images are really small to appreciate the lack of colocalization between the markers and thus the lack of expression of GPR158 in those cells. Primary cultures could be prepared to asses the lack of expression of GPR158 in glia cells.
Figure 7 legend should be complemented to detail each diagram.
Reviewer 2 Report
This is a great article on GPR158. Can the authors elaborate on whether it is also proton-sensitive? Other GPCRs are altered in terms of their activity via extracxellular pH changes.
Reviewer 3 Report
I have received the manuscript entitled “Expression mapping and functional analysis of orphan G-protein coupled receptor GPR158 in the adult mouse brain using a GPR158 reporter mouse” by Jinlong Chang et al. for evaluation. The authors have investigated the localization, expression and function of GPR158 linked with various neurological diseases of central nervous system. They have also performed the transcriptomic and proteomic analysis in GPR158Tag and GPR158-/- mice.
The study is novel, well planned and very impressive therefore I have very few comments regarding this study. My comments and critics are given below:
Major comments:
1- Authors have performed most of the experiments in GPR158Tag except RNA-seq experiment. It is worthwhile and essential to know whether the GPR158-/- mice behaves differently.
2- In the fig.2 F, G & H, authors have investigated the sociability and social novelty behavior in GPR158Tag mice. In result section, authors have claimed that there is no social impairment and social novelty deficits in GPR158Tag mice compared to control mice. But its looks from the bar diagrams that GPR158Tag mice are more social than Wild type mice therefore I would suggest authors to calculate the ratio of Empty/ S1in fig G and S2/S1 in fig H.
3- Further I would suggest authors to perform the sociability and social novelty behavior in GPR158-/- mice.
Minor comments:
1- Authors have mentioned ‘Maker’ word throughout the manuscript I hope it is ‘marker’ line no. 245. If it is true, please correct it if not explain its meaning.
2- In the methodology section, Authors have not cited the any reference in many subsections including 2.4 Three-chamber social interaction test, 2.5 Novel Object Recognition Test and 2.6 Immunofluorescence staining. Please cite the appropriate references.
3- In fig2 & 3, the font size of text is either very small or too much crowded. Please improve it for clear visibility.
Reviewer 4 Report
The following points may be of concern at a quick view
Line 253: components of tissue lysis solution?
Also many other buffers are not properly described…
Line 273: Why select only bands that are near the molecular weight of GPR158 in a denaturing SDS PAGE? Interactors may also have completely different molecular weights
In Section 3.7 and Figure 7: No information is provided about the reproducibility of candidate interactor protein detection over multiple replicates and negative controls (antibody isotype control; no tag etc.). How can non-specific interactors be distinguished that only bind to the beads or are off targets of the antibody.
Round 2
Reviewer 3 Report
Accepted
Author Response
Thank you for your valuable comment.
